# A novel cluster-based learning scheme to design optimal networks for atmospheric greenhouse gas monitoring (CRO<sup>2</sup>A version 1.0)

David Matajira-Rueda<sup>1</sup>, Charbel Abdallah<sup>1</sup>, and Thomas Lauvaux<sup>1</sup>

<sup>1</sup>Groupe de Spectrométrie Moléculaire et Atmosphérique GSMA, UMR CNRS 7331, Université de Reims-Champagne Ardenne, UFR Sciences Exactes et Naturelles, Moulin de la Housse, BP 1039, 51687, Reims Cedex 2, France Correspondence: David Matajira-Rueda (david.matajira-rueda@univ-reims.fr)

Abstract. With the continued deployment of atmospheric greenhouse gas monitoring networks worldwide, optimal and strategic positioning of ground stations is essential to minimize network size while ensuring robust observation of fossil fuel emissions in large and diverse environments. In this study, a novel scheme (Concepteur de Réseaux Optimaux d'Observations Atmosphériques - CRO<sup>2</sup>A) is developed to design optimal mesoscale atmospheric greenhouse gas monitoring networks through a three-stage process of unsupervised clustering with inverse weighting and data processing. Unlike current approaches that rely primarily on inverse-modeling pseudo-data and heavily on error or uncertainty assumptions, this scheme requires no such assumptions; instead, it relies solely on direct atmospheric simulations of greenhouse gas concentrations. The CRO<sup>2</sup>A design scheme improves convergence to an optimal solution by minimizing the number of ground-based monitoring stations in the network while maximizing overall network performance. It can perform both foreground and background analyses and can assess and diagnose the quality of existing monitoring networks, among other special features. CRO<sup>2</sup>A treats simulated greenhouse gas concentration fields as spatiotemporal images, processed through multiple transformations, including data cleaning and automatic information extraction. These transformations reduce processing time and sensitivity to outliers and noise. The developed scheme incorporates techniques such as image processing and pattern recognition, supported by optimal heuristics derived from operations research, which enhance the ability to explore and exploit the problem search space during the solution process. Two applications are presented to illustrate the capabilities of the proposed optimal design scheme. These are based on simulations of atmospheric CO2 concentrations from the Weather Research and Forecasting (WRF) model-one for an urban setting and the other for a regional case in eastern France-used to evaluate optimal network designs and the computational performance of the scheme. The results demonstrate that the design scheme is competitive, straightforward, and capable of solving the design problem while maintaining a balanced computational cost. Based on the WRF reference simulation, CRO<sup>2</sup>A performed analyses of foreground measurements (atmospheric signatures of fossil fuel emissions) and their associated background fields (where simulated large-scale background concentrations are used, avoiding major sources and sinks of greenhouse gases), providing the minimum number of ground-based measurement stations and their optimal locations in the regions. As additional features, CRO<sup>2</sup>A enables users to diagnose the performance of any existing network and improve it in the event of future expansion plans. Furthermore, it can be used to design and deploy an optimal monitoring network based on predefined potential locations within the region under analysis.

#### 1 Introduction

Atmospheric inversions of greenhouse gas (GHG) emissions have been applied at the global scale to better understand the spatial and temporal distribution of sources and sinks across continents and oceans (Enting and Mansbridge, 1989; Bousquet et al., 2000). At finer scales, denser observation networks have been deployed, such as the atmospheric network of the Integrated Carbon Observing System (ICOS) <sup>1</sup>, the permanent National Oceanic and Atmospheric Administration (NOAA) tall tower network (Andrews et al., 2014), and temporary regional networks (e.g., the Mid Continent Intensive Experiment (Miles et al., 2012)). More recently, urban networks of ground-based stations have been established in Indianapolis (Miles et al., 2017), Paris (Doc et al., 2024), Los Angeles (Kort et al., 2013), and Mexico City (Taquet et al., 2024). These networks were designed to address specific questions, such as regional carbon budgets (Lauvaux et al., 2012), ecosystem responses to climate variability (van der Woude et al., 2023), or long-term trends in fossil fuel emissions (Lian et al., 2023). However, their design is often guided by non-objective criteria based on expert judgement, aiming to capture atmospheric signatures of GHG fluxes from biogenic or anthropogenic sources and sinks. Beyond the practical requirements of specific instrumentation (e.g., pre-existing infrastructures, temperature-controlled environments, absence of nearby sources, etc.), optimal network design remains a key research question. This requires objective criteria to address specific scientific goals while limiting the number of measurement locations and improving network performance.

Many studies have proposed algorithms for designing observation networks that reduce uncertainties in estimating GHGs (CO<sub>2</sub>, CH<sub>4</sub>, and N<sub>2</sub>O) at the global scale (Rayner et al., 1996) and in specific regions, such as the African continent (Nickless et al., 2020), North America (Shiga et al., 2013), Northern Europe (Super et al., 2020), Italy (Villalobos et al., 2025), or China (Wang et al., 2023). Common approaches include the use of Bayesian inversion models, atmospheric Eulerian transport models, or Lagrangian Particle Dispersion Models (LPDMs) to describe the influence areas for each measurement location (Nalini et al., 2019; Sim et al., 2024), combined with multi-objective optimization algorithms such as Incremental Optimization (Nickless et al., 2020), Genetic Algorithms (Lucas et al., 2015), or Simulated Annealing (Patra and Maksyutov, 2002). In most schemes proposed, objective functions consider a range of factors-geographical, seasonal, spatio-temporal, physical, and economic (Hari et al., 2016).

Patra and Maksyutov (2002) proposed an inversion model-based design approach using Incremental Optimization (compared with Simulated Annealing) for global observational networks, aiming to maximize constraints on CO<sub>2</sub> flux uncertainty using simulations from the Semi-Lagrangian NIES (National Institute for Environmental Studies, Tsukuba)/FRSGC global transport model. Ziehn et al. (2014b) employed an inverse-mode Lagrangian particle dispersion model coupled with a Bayesian inversion framework, using simulated fields from the regional version of the Australian Community and Earth System Simulator (ACCESS), and found negligible influence from data outside the domain. The optimization method in the study by Ziehn et al. (2016) was incremental and multi-objective, accounting for establishment and maintenance costs. They observed that uncertainty reduction is maximized when designing a monitoring network without considering the existing network, and noted a saturation effect when increasing the number of observing stations beyond a certain point.

<sup>1</sup>https://www.icos-cp.eu



Lucas et al. (2015) developed a method to design an optimal GHG observing network that incorporates several factors beyond simulated data from the Weather Research and Forecasting Model with Coupled Chemistry (WRF-Chem) for the target GHG. Their approach used a Bayesian inversion model and a multi-objective genetic optimization algorithm in an incremental scheme to minimize both emission uncertainties and station measurement costs. Given that urban areas account for about 70% of anthropogenic CO<sub>2</sub> emissions, several studies have attempted to design optimal urban networks to assess the effectiveness of emission reduction strategies (e.g., Vardag and Maiwald (2024)). Shusterman et al. (2016) sought to balance the quality and quantity of low-cost observing stations in Oakland, California, and surrounding areas for the Berkeley Atmospheric CO<sub>2</sub> Network (BEACO2N), using cost, reliability, accuracy, and systematic uncertainty to characterize their network. The close relationship between design and optimization is most frequently applied in engineering and often requires novel strategies supported by high-performance computational resources. Thompson and Pisso (2023) proposed a scheme for designing atmospheric monitoring networks based primarily on information theory-that is, statistical data processing-without employing the uncertainty reduction approach. This scheme can handle various types of atmospheric data and is computationally efficient, as it does not require repeated inversion of large matrices. However, it is constrained by the dimensions of certain matrices, which can lead to hardware memory overloads.

The use of an inversion model is computationally expensive and often subject to restrictions and assumptions that limit its accuracy and reliability-factors already affected by the model implicit regularization process. The assumptions made when defining prior error covariances frequently rely on subjective definitions, which can guide the optimization process and inter-station distances more than the atmospheric transport itself (Lauvaux et al., 2012). Furthermore, a successful application requires reframing the ill-posed problem as an approximation of a well-posed problem; such reframing must satisfy the well-known Hadamard conditions (Shiga et al., 2013; Hadamard, 1902). Therefore, given recent advances in data analysis and processing, new solution techniques are emerging, offering alternatives to classical methods and representing a potential paradigm shift (e.g., Lopez-Coto et al. (2017)). The approach proposed in our study (*Concepteur de Réseaux Optimaux d'Observations Atmosphériques* – CRO<sup>2</sup>A) not only seeks to avoid the inversion process and to evaluate alternative solutions, but also aims to change the problem formulation itself. The core concept is to observe and analyze trends in data simulated by transport models, selecting ground-based monitoring station locations that ensure the presence of significant GHG concentrations for the majority of the simulation period.

For the sake of the extension of our proposal, and since it is described in detail in Sect. 2.2 and its foundations are presented in Appendix. B and C, it only remains to mention some of the motivations about machine learning and clustering algorithms that gave the basis for the development of this project.

For machine learning algorithms, several factors guide the appropriate choice, including prevention of overfitting and underfitting, data characteristics, and, critically, understanding the problem. Understanding the problem depends on recognizing its defining characteristics. In pattern recognition, when the training dataset consists solely of input values without corresponding outputs, the problem is typically addressed using unsupervised learning algorithms. These algorithms can reveal clusters of similar (or dissimilar) data points-referred to as clustering-which is equivalent to estimating the data point distribution (i.e., density estimation) (de Burgh-Day and Leeuwenburg, 2023).




Clustering algorithms commonly model similarity (or dissimilarity) using distance metrics (i.e., the distance paradigm). Depending on the chosen metric, substantially different results may be obtained according to the clustering criteria applied (Wang and Song, 2011; Doan et al., 2022). Distance metrics are widely used due to their ease of implementation and broad acceptance within the scientific community. However, recent studies indicate they are less effective when handling noisy or irrelevant data points. Feature weighting has been suggested as a potential solution to this limitation, which becomes particularly pronounced for temporal or spatial series (Doan et al., 2022).

Given the problem characteristics, it is impossible to fully map the search space, and since the number of possible solutions increases factorially with the number of clusters, it is necessary to conduct an initial global search (exploration) followed by a local search (exploitation) of candidate solutions in the aforementioned space. The success of this search depends on the heuristic strategy adopted and on the choice or definition of the objective (cost) function.

Based on the above, the main characteristics of the data used in this research are presented in Sect. 2.1. The applied methodology is then described in detail-almost as a user manual-through the three stages of the optimal design scheme in Sects. 2.2.1, 2.2.2, and 2.2.3. The results demonstrate the potential of the CRO<sup>2</sup>A scheme in two applications-at the urban scale and at the regional scale-both in northeastern France, focusing on quantifying fossil fuel CO<sub>2</sub> emissions (Sect. 3). For each application, CRO<sup>2</sup>A is used to: first (foreground analysis) explore the design of a network to monitor fossil fuel CO<sub>2</sub> emission signatures, and second (background analysis) address the need for measurements capable of capturing large-scale CO<sub>2</sub> input concentrations (background observations). Discussions and conclusions are provided in Sects. 4 and 5, respectively. Finally, special features of the optimal design scheme are summarized in Appendix A, technical notes on the clustering process underlying the CRO<sup>2</sup>A development are given in Appendix B, and additional and complementary proofs of results are provided in Appendix D.

#### 2 Methods and materials

## 115 2.1 Modeling and simulation of atmospheric CO2 concentrations

The data used in this study were simulated using the Weather Research and Forecasting model (Skamarock et al., 2008; Guo et al., 2024; Debnath et al., 2024) with the passive tracer transport option of its chemistry module (WRF-Chem v3.9.1) to generate  $CO_2$  concentration fields. For the regional application, the WRF-Chem domain was centered over the region of interest and extended beyond its boundaries toward Paris and neighboring countries. The model grid comprised 50 vertical levels and a horizontal resolution of  $3 \times 3$  km, resulting in  $201 \times 150$  grid points per level. For lateral boundary conditions, we used European Centre for Medium-Range Weather Forecasts (ECMWF) Reanalysis v5 (ERA5) hourly meteorological data (Copernicus Climate Change Service, 2018) and Copernicus Atmosphere Monitoring Service (CAMS) global inversion-optimized greenhouse gas concentrations for  $CO_2$  (Copernicus Atmosphere Monitoring Service, 2024). Anthropogenic fluxes were extracted from the high-resolution (1 × 1 km) TNO dataset for 2019 (van der Gon et al., 2019), while biogenic fluxes were modelled using the Vegetation Photosynthesis and Respiration Model (VPRM (Mahadevan et al., 2008)). To distinguish between  $CO_2$  sources, we separated total concentrations into three components: anthropogenic, biogenic, and background.

The anthropogenic signal was further decomposed to separate  $CO_2$  originating from the Grand Est region from that of the rest of the domain. For the urban application, the model configuration was identical except for the resolution (1 × 1 km), fully nested within the parent domain used for the regional application.

# 130 2.2 The CRO<sup>2</sup>A optimal design scheme

The processing performed by the three main stages of the  $CRO^2A$  (version 1.0) optimal scheme is described below, detailing each transformation applied to the data. Throughout these stages, an illustrative example is presented, assuming seven ground-based measurement stations (k = 7) within a masked version of the Grand Est region (northeastern France) to show both the inputs and outputs of each transformation. The proposed scheme is represented as a MISO (Multiple-Input, Single-Output) system in Fig. 1, with its core based entirely on an inverse-weighted clustering process.

# 2.2.1 Pre-processing





This initial stage treats the data as images to extract the maximum amount of useful information after applying cleaning, filtering, and masking processes (i.e., image processing techniques). As input, the CRO<sup>2</sup>A pre-processing stage primarily requires simulated spatio-temporal datasets (see Fig. 2a and 2b) containing simulated greenhouse gas concentrations for a specific region (see Fig. 2c), generated using the resources described in Sect. 2.1.

The following description corresponds to the sequence of procedures within the yellow dashed line box in the flow chart of Fig. 1. The first transformations are applied sequentially to reveal relevant information and aid in the data cleaning process.

The variables of the selected tracer (the greenhouse gas to be analyzed) and the corresponding latitude and longitude coordinates delimiting the study region are defined and stored in matrices projected according to the Coordinate Reference System (CRS), here WGS84/pseudo-mercator.

In CRO<sup>2</sup>A, it is possible to select a subregion for analysis (i.e., an area within the boundaries of the input dataset without recompilation). This can be done graphically or manually by specifying only the minimum and maximum coordinate quadrants of the latitudes and longitudes.

Tracer data are converted into images in line with the proposed approach to leverage this representation. This is equivalent to treating the data as matrices allowing fast indexing, where each element specifies the color of a pixel in the image.

The row and column indices of the elements determine the centers of the corresponding pixels and, in turn, the associated concentration value. The converted datasets are stored in a three-dimensional array whose dimensions define the image height and width and the number of images equals the length of the time vector. For example, from a  $(150 \times 201 \times 768)$  dataset, 768 frames or images (equal to the time vector length) can be obtained, as shown in Fig. 2a, each with  $150 \times 201$  pixels (height and width, respectively).

In the pseudo-color plots, the logarithmic values of each matrix from the arrays are displayed as colored flat surfaces in the x-y plane.

This procedure can be used for visualization and verification; however, each matrix can also be directly converted to a grayscale image without first converting it to a true-color RGB (Red-Green-Blue) image.

**Figure 1.** Schematic flow chart of CRO<sup>2</sup>A illustrating the proposed scheme for designing optimal networks for monitoring atmospheric greenhouse gases. The inversely weighted clustering-based learning scheme (black dashed line box) is detailed in three stages of development: pre-processing (green dashed line box), processing (yellow dashed line box), and post-processing (red dashed line box).

This approach reduces storage requirements, as only one channel (grayscale) is needed instead of three (RGB) to represent the same information. In addition to storage savings, this method can significantly reduce the processing time of the proposed design scheme and ensures that the result is independent of the color space chosen for graphical representation. For these reasons, every matrix in the array is converted to a grayscale image, as shown in Fig. 3a, with values in the range [0,1]-corresponding to pixels from black to white, respectively.

**Figure 2.** First pseudo-color image of the simulated atmospheric CO<sub>2</sub> concentration dataset over the analyzed region at 50 m agl (a), its three-dimensional surface with a contour plot underneath (b), and its corresponding geographical region (northeastern France) delimited by the black dashed line (c).

Once this transformation is applied to each image, a datastore is created to hold the entire collection. The datastore enables rapid processing without exceeding available memory, particularly when handling a large number of images. A further advantage of using datastores is that transformations can be applied to the entire collection simultaneously rather than individually in sequence.

To isolate pixels with intensities close to the target concentrations-defined by an instrument sensitivity or through contour analysis (as described below)-it is recommended to adjust image contrast by uniformly modifying pixel intensity values according to their magnitudes. This adjustment shifts them toward either the bright or dark range, depending on the data and user-defined target, and is achieved through contrast stretching. Two contrast adjustment options are proposed: (i) logarithmic rescaling of pixel intensities according to concentration magnitudes (see Fig. 3b), and (ii) histogram equalization (see Fig. 3c).

In both cases, an optimal thresholding process is required, which can be considered a separate topic of study. The advantages and disadvantages of each method depend on the concentration values of interest, the size of the resulting dataset, and the associated processing time. Typically, the equalized version includes a larger proportion of the data, resulting in longer processing times than the scaled version.

190

Figure 3. Contrast adjustment of the grayscale version of Fig. 2a (a), its logarithmically rescaled version (b), and its equalized version (c).

The differences between the two options are shown in Fig. 4, which presents the histogram transformations for versions of the same processed image. These figures highlight an implicit duality between the two transformations. Brighter pixels (intensity values close to 1) are more prominent in the equalized version (see Fig. 4c) than in the rescaled version (see Fig. 4b). Such pixels are the focus of this procedure, though it is important to note that revealing more pixels increases the data volume and processing requirements.

In this case, by default, contrast adjustment is performed by logarithmic rescaling, using the value obtained from contour analysis of the concentration magnitudes in each image as a threshold. Once the grayscale image contrast is adjusted, the image is binarized (see Fig. 5a) using the well-known Otsu method (Liu and Yu, 2009; Bangare et al., 2015), which determines an optimal threshold corresponding to the minimal intraclass variance from a 256-bin histogram. Pixels with intensities above the global threshold are set to 1 (white), and those below are set to 0 (black). Both contrast adjustment and binarization define the search (solution) space for each image. These transformations isolate the information of interest-white pixel regions with intensity 1 in Fig. 5a-so that only target GHG concentrations are analyzed, while remaining areas are ignored as irrelevant or noise.

Once the binarized image collection is obtained, masking is applied to exclude all bodies of water or other undesired areas (when visible at the given resolution and dimensions), as these locations should not be part of the search space. In the illustrative example, the mask in Fig. 5b is used to analyze only CO<sub>2</sub> concentrations within the Grand-Est region.

**Figure 4.** Histograms of the contrast adjustment of Fig. 2a (a), the compressed version by logarithmic rescaling (b), and the uniformly distributed (decompressed) version by equalization (c).

Figure 5. Binarized version of Fig. 3b (a) and the mask applied to every image (b).

Thus, the solution space for each image, initially defined through earlier transformations, is further restricted to account for the topographic conditions of the study area. The masking procedure uses an equally binarized image with dimensions matching those of the dataset. Leveraging the binary values (1 and 0) that correspond to Boolean true and false, a logical AND operation is performed via a dot product between each image and the mask. This operation is inherent to computing systems and is computationally inexpensive compared with alternative methods.

As one of the final pre-processing steps, all resulting binarized images are accumulated (superimposed), producing a scoring matrix whose elements are the sums of pixel intensities at each position. The scoring matrix represents the spatial distribution of significant concentrations, expressed as the frequency of occurrence at each location. For example, for annual simulations (365 days), a matrix element with a value of 75 indicates that the location exhibits substantial GHG concentrations for 75% of the simulation period (approximately 273 days). The scoring matrix (Fig. 6) depends directly on the pre-processed data and is used in the clustering process for weighting during both processing and post-processing stages.

**Figure 6.** Scoring matrix of concentrations (from 0% to 100%) representing the frequency of substantial concentration enhancements across the region.

Finally, before initiating the processing stage, the reference raster postings to geographic coordinates and the datastores are saved. The distance metric (squared Euclidean distance by default), maximum number of iterations (ite), repetitions (run), replicates (rep), and saturation value (sat) are also defined; some of these serve as stopping criteria for decision-making and optimization steps. Their use is described later.

## 210 **2.2.2 Processing**

205

This intermediate iterative stage aims to identify similar or dissimilar groups by clustering the pixels detected in each image from the previous stage. The resulting groups are characterized according to the patterns that generated them (i.e., pattern recognition techniques are applied to the data). The following description corresponds to the sequence of procedures within the blue dashed box in the flow chart shown in Fig. 1.

The CRO $^2$ A processing stage requires the output images from the pre-processing stage (see Fig. 5a), the scoring matrix (see Fig. 6), and the number of clusters under consideration (k), corresponding to the number of ground-based measurement stations.

235

Using the binarized images, the transformed dataset identifies pixels with an intensity of 1 (colored and marked pixels in Fig. 7). For each image, these data points define the search space for the solution, as they represent the locations of the relevant concentrations (targets).

To apply the clustering process to these data points, the clustering algorithms must be initialized by defining starting points (i.e., a priori positions of ground-based measurement stations) that act as initial centroids in iteration zero. This initialization is commonly performed by selecting k points uniformly distributed within the search space, representing the simplest among several possible options.

Each clustering problem constitutes an optimization problem, and the present development assumes a global solution in both search space and values. To reduce the risk of converging to a local minimum, and given that the underlying algorithms (*k*-means and *k*-medoids) are sensitive to initial conditions, the starting points for clustering are defined using information from the marginal distributions of each image width (horizontal axis) and height (vertical axis). These are obtained as univariate histograms (see Fig. 7). These marginal distributions show the vertical and horizontal densities of pixels (i.e., data points) per row and column, respectively.

Figure 7. Univariate histogram (marginal distributions) of Fig. 5a used to define the starting points of the clustering.

This considerably reduces processing time and the number of iterations, as the starting points are typically located close to the highest concentrations. This positioning guides the search vectors during solution exploration, thereby requiring fewer iterations to converge to an optimal solution. The size of the starting point matrix is  $k \times 2$ , determined by the number of clusters tested (k) and the number of features-in this case, 2, corresponding to latitude (y-projection) and longitude (x-projection) for each point. In other words, the number of starting points equals the number of ground-based measurement stations in the monitoring network.

Once the pixels are identified, a primary grid is superimposed on the image, with the number of divisions in both height and width equal to k, resulting in  $k^2$  cells (see vertical and horizontal black lines in Fig. 8). For each division in width and height, the maximum value of the corresponding portion of the marginal distribution is selected. A secondary grid is then drawn, representing the locations most likely to host ground-based measurement stations (intersections of the green dashed lines in Fig. 8).

Depending on the image characteristics, up to  $(k-1)^2$  starting point candidates can be obtained (i.e., the maximum number of internal intersections between the horizontal and vertical grid lines). However, even though the theoretical maximum is  $(k-1)^2$ , some candidates are discarded if they do not belong to the search space (i.e., identified pixels). In some cases, certain cells may contain no identified pixels from the previous procedure, reducing the number of available starting points to less than  $(k-1)^2$ .

If the number of available candidates is greater than or equal to k, exactly k are randomly selected and ordered. To diversify the clustering process, a starting point array of size  $k \times 2 \times k$  can be created by assigning random permutations of the selected points. This strategy allows the clustering process to be replicated with a different ordering of starting points, to which the clustering algorithms are also sensitive.

If the number of candidates is less than k, those options are discarded and the starting points are instead selected uniformly at random within the search space.

**Figure 8.** Primary grid corresponding to the first binarized and masked image of the dataset (see Fig. 7) (black lines), the overlapping marginal distributions for each row (red) and column (blue), and maximal values of the marginal distributions (green) for each vertical and horizontal division (secondary grid). Starting point candidates are at green line intersections; selected starting points are shown as black circles.

Since the number of candidates for the first analyzed image in the illustrative example is greater than k = 7, the randomly selected starting points are shown as black circles in Fig. 8. Each selected starting point can be verified to belong to the identified pixels.

Before starting the clustering process, if any ground-based measurement station is already installed in the analyzed region and the user wishes to include it, its location can be added to the dataset by applying a multiplicity concatenation equivalent to 2.5% of the total number of images. The clustering algorithm then partitions the dataset into k clusters and returns: a vector containing the cluster index of each data point; the k cluster centroid locations; the within-cluster sums of point-to-centroid distances; and the distances from each point to every centroid. In other words, the clustering process characterizes the data points identified in each image according to patterns defined by the distance relationships between each data point and the centroids. Such characterization can be used to identify the most appropriate cluster for each point, considering all possible clusterings that meet an optimal criterion. The optimal criterion is based on the distance between a pixel and its cluster centroid, the distances between centroids, and the scoring matrix value (weight) at the pixel location.

This stage operates under the hypothesis that the location of the ground-based measurement station (the cluster centroid) should be as close as possible to the source of greenhouse gases (highest concentration) to ensure accurate measurements and improve subsequent data inversion, if required. This is the same hypothesis used to define the clustering starting points. The gradual formation of the seven clusters is shown through the changing colored groups of identified pixels.

**Figure 9.** Data points associated with the different clusters and their corresponding centroids (black triangles) for the first processed image (a) and the collection of centroids resulting from processing all images in the dataset (b).

In Fig. 9a, the processing results for the first image in the dataset are shown after 27 iterations. The seven clusters obtained (k = 7) are displayed in different colors. The x-y projection of each cluster centroid is indicated by black triangles, representing the optimal locations for ground-based measurement stations according to the first image (see Fig. 9a). The relationship between the randomly selected starting points and the pseudo-global centroids confirms the hypothesis of spatial closeness and convergence proposed through the use of marginal distributions. It should be noted that the optimal clustering is obtained without requiring clusters to contain the same number of members; instead, it depends on the distances between centroids and members, as well as the scoring matrix values at the member locations.

Maximizing inter-centroid distances produces well-separated clusters that can form, expand, contract, and exchange members dynamically. Centroid tendencies align with regions of highest greenhouse gas concentration and correspond to dense pixel areas in the scoring matrix.

In this case, initializing starting points using the proposed strategy results in a low number of iterations to reach an optimal solution, as outcomes tend to follow the stated hypothesis. Convergence is thus relatively fast, reducing computational cost and reinforcing the hypothesis. As a result of the processing stage, after clustering each image in the dataset, the resulting centroids (see Fig. 9b), referred to as local centroids, are stored for use in the post-processing stage. The centroids shown for the first image in Fig. 9a are elements of the cluster sets in Fig. 9b, or the set of local centroids. In total, 5,376 local centroids are identified in Fig. 9b, as the illustrative example comprises 768 frames with k = 7 clusters each.

#### 285 2.2.3 Post-processing



This final stage performs a general cluster analysis, estimates clusters, determines centroids, and displays optimal results. In addition to pattern recognition techniques, optimization and operations research methods are applied to the results collected in the processing stage. The following description refers to the sequence of procedures within the dashed red box in the flow chart in Fig. 1.

As input, the CRO<sup>2</sup>A post-processing stage requires the output data from the processing stage (local centroids of the images in Fig. 9b) and, as in the processing stage, the scoring matrix and the number of clusters to be evaluated (k). Although the processed data from both stages appear similar, they represent different information. In the processing stage, the spatio-temporal distributions of GHG concentrations were used (sets of images such as those shown in Fig. 5a). In the post-processing stage, the collection of local centroids from each image is used (a single image as shown in Fig. 9b).

Building on the learning from the processing stage, the cluster analysis here again considers the large dataset size; however, it now processes representative entities (local centroids) rather than individual data points analyzed previously.

Once the overall results of the new clustering process are obtained (global centroids; see Fig. 10a), i.e., after clustering the data from the processing stage, percentage scores for each ground-based measurement station location and for the monitoring network are calculated using the scoring matrix, as shown in Fig. 10b. Promising permutations of the global centroids are then used to further explore the search space. Alternative solutions are tested around the general solution, i.e., reclustering with different permuted and modified starting points. This approach is analogous to forcing the Pareto profile in a multi-objective optimization, but at low computational cost. After at least 25 repetitions, to ensure adequate statistical reliability, the clustering with the highest mean score for a specific value of k is considered the optimal solution (k\*), provided it outperforms the general solution; otherwise, the general solution is retained due to the lack of improvement.

Finally, CRO<sup>2</sup>A outputs the geographic coordinates of the  $k^*$  ground-based measurement stations, their individual scores, and the score of the optimally selected monitoring network.

It should be noted that in clustering algorithms (both k-medoids and k-means), one of their characteristics is that the number of k clusters used to represent the dataset is predefined before the algorithm starts.

Figure 10. Resulting optimal centroids (black triangles) for all images in the dataset (a), and their corresponding score percentages (b).

Consequently, the algorithm iteratively modifies the locations of the centroids of these predefined clusters k, based on the calculation of the mean, which reflects the density distribution of the dataset. Given this characteristic, a trend analysis is implemented using the basic sequential algorithmic scheme (BSAS) (Theodoridis and Koutroumbas, 2006). This approach is proposed to address the persistent and subjective question of the minimum number of clusters required for adequate clustering (i.e., the number of ground-based measurement stations in the network and their corresponding locations in the region under analysis).

Through this analysis and batch processing, as illustrated in the first sub-Fig. 11, an interval of clusters is tested sequentially and incrementally (in the example, [2,15]), providing information on the overall quality of the monitoring network as a function of the number of ground-based measurement stations.

Figure 11. Logistic fit (S-shaped curve) of the network performance as a function of monitoring tower numbers.






The results from this analysis are fitted to a sigmoidal curve, and the method presented by McDowall and Dampney (2006) is applied to determine the optimal threshold. This threshold indicates the minimum number of ground-based measurement stations required for the monitoring network (see Fig. 11). The sigmoidal function selected in this case is particularly useful for modeling artificial neural networks. It typically describes complex systems characterized by learning curves with rapidly ascending intermediate transitions from low levels to a saturation point at high levels after a significant increase in the independent variable. A specific case of the sigmoidal function is the logistic function, widely used in regression models. The logistic function and its first derivative are employed to calculate the optimal threshold (see the second sub-Fig. 11).

According to Fig. 11, the illustrative example developed through these three stages shows that the minimum number of monitoring towers for the masked region corresponds to the calculated threshold. The optimal value for the overall quality of the network is approximately 57.14%, as shown in Fig. 10b. As presented in these sections, complete processing of these large datasets is achieved through the training and validation of a machine learning system capable of generating optimal designs for atmospheric greenhouse gas monitoring networks based on such datasets. This unsupervised machine learning scheme adapts proportionally and progressively to the amount of data available during processing, improving its performance even under uncertainty.

As noted at the outset, CRO<sup>2</sup>A is structured in three stages, each functioning as an analysis module. Partial results are saved after each stage, allowing processing to be paused and resumed later. It is also important to note that once pre-processing is completed, it need not be repeated for either a single monitoring tower configuration or for trend analysis.

Finally, Fig. 12 presents the algorithmic complexity analysis of processing times for the three stages of the  $CRO^2A$  design scheme. Figure 12a shows the processing times as a function of the number of images in the pre-processing stage and their corresponding curve fitting. The linear relationship between these variables is notable. Despite this characteristic, for datasets with larger numbers of images, the pre-processing stage does not impose a higher computational cost due to the use of datastores. Figure 12b shows the processing times for all three stages as a function of the number of clusters (or ground-based measurement stations). The processing times for the pre-processing stage remain constant, as they are independent of the number of clusters (k). The most evident variations occur for 2 

For each, the results of the trend, foreground, and background analyses are shown. To limit computational resource requirements, two one-month simulations for October 2022 and 2023 were used. While this constrains the absolute representativeness of the proposed networks, the simulations are sufficient to demonstrate the applicability and utility of the design scheme.





Figure 12. Algorithmic complexity analysis of processing times for the three stages of the CRO<sup>2</sup>A design scheme.

Designing an operational network would require longer simulations encompassing all meteorological conditions across the four seasons. First, for the foreground analysis, the scheme described in the previous section is applied. The target tracer (the greenhouse gas variable to be analyzed) is selected, along with the interval for the number of monitoring towers to be tested. Second, for the background analysis, the scheme is applied directly to the background tracer using the optimal value obtained from the foreground analysis, with the special masking mode activated. This mode selects the tracer used in the foreground analysis, restricting the search space for the background tracer to locations with the highest greenhouse gas concentrations. The details of the special masking feature are provided in Appendix A. Third, the results from both the foreground and background analyses are compiled, and the corresponding figures and coordinates of the elements comprising each designed network are presented.

The background analysis uses the same number of monitoring towers as determined in the foreground analysis (i.e., creating a one-to-one network). However, at expert discretion, some of the ground-based measurement stations proposed in the background analysis may be removed following appropriate evaluation, primarily based on their separation distances.

# 3.1 Urban-scale application

For the first application, the CRO<sup>2</sup>A design scheme was applied to the urban area of Reims (France, Grand Est region), using atmospheric  $CO_2$  concentrations simulated by WRF-Chem at 1 km resolution. The fossil fuel component (individual tracer) was extracted from the simulation to isolate the urban fossil fuel signal, while the background analysis included the large-scale inflow and local biogenic fluxes. Fig. 13 shows the logistic fitting curve of the trend analysis results at the urban level. The tested range for the number of monitoring towers was [2,10] with a unit step. The resulting threshold represents the minimum number of ground-based measurement stations required for the region under study (in this case, at least three according to  $CRO^2A$ ). Figures 14a and 14b show the resulting clusters and their corresponding global centroids for a network composed of three ground-based measurement stations (k = 3) for the foreground and background networks, respectively.

Similarly, Fig. 14c and 14d show the locations of the optimal centroids relative to the emissions field for the analyzed region at the urban scale and the scoring matrix. Pre-existing towers were not included in the optimization procedure.


A similar analysis, including them (black triangles in Fig. A1a and A1b), is presented in Appendix A1. The three foreground locations correspond to (i) the downtown urban area, (ii) the sugar factory located to the northeast of the city, and (iii) a site to the northwest where both the city plume from Reims and the sugar factory plume converge. When a fourth measurement location is added (Appendix D), it is positioned to the southeast of the city near a smaller sugar factory, enabling separation of the urban plume from the second sugar factory plume.

Background tower locations, identified by proximity and low scoring matrix values, are situated to the west of the city-where  $\mathrm{CO}_2$  enhancements are minimal-and upwind of the two sugar factories. When the two pre-existing ground-based measurement stations are considered, the optimal network size increases to k=5, indicating that the current tower locations are sub-optimal (outside the main city plume). However, a longer simulation period would be required to draw conclusions regarding the overall effectiveness of the current towers in measuring urban plumes. The coordinates of the optimal ground-based measurement station locations (urban scale) and their corresponding performance values are provided in Table 1.

Figure 13. Logistic fit (S-shaped curve) of the urban network performance as a function of monitoring tower numbers.

Table 1. Optimal results coordinates for urban-scale analysis.

|   | ]         | Foreground | I         | Background |          |           |
|---|-----------|------------|-----------|------------|----------|-----------|
|   | Longitude | Latitude   | Score (%) | Longitude  | Latitude | Score (%) |
| 1 | 4.0358    | 49.240     | 60.101    | 4.2177     | 49.326   | 54.057    |
| 2 | 3.9099    | 49.371     | 28.100    | 4.1617     | 49.149   | 58.921    |
| 3 | 4.1524    | 49.368     | 42.042    | 3.8167     | 49.240   | 58.777    |

**Figure 14.** Resulting optimal centroids (black triangles) and their corresponding clusters for all images in the dataset for both the foreground and background networks (urban level) (a), location of the optimal centroids relative to the emission field (b), and the scoring matrix (c).

**Figure 15.** Binarized and inverted image of the score matrix used to mask the background tracer in the background analysis for the urban-scale application.

# 3.2 Regional-scale application

For the second application, a regional tower network in northeastern France was considered, centered on the *Grand Est* region, which is dominated by croplands and forests and contains several large metropolitan areas, including Paris, Strasbourg, Nancy, Metz, and Reims in France; Frankfurt and Karlsruhe in Germany; and Basel and Zurich in Switzerland.




The study domain also includes several industrial areas, such as car manufacturing and highway traffic in Alsace, and large industries in the Ruhr Valley along the Rhine River, which produce noticeable atmospheric CO<sub>2</sub> plumes. The WRF-Chem model configuration, described in Sect. 2.1, was run at 3 km resolution for two months (October 2022 and 2023). Results excluding the existing ICOS tower network are presented here, while the optimal network including the current ICOS station locations is shown in Appendix D. As in the urban-scale application, pre-existing towers were not included in the optimization procedure. A similar analysis that includes them is presented in Appendix A2 (black triangles in Fig. A2a and A2b).

Figure 16 shows the logistic fit of the trend analysis results over the regional domain. The calculated threshold represents the minimum number of ground-based measurement stations required for the region under study (in this case, at least nine according to CRO<sup>2</sup>A processing). Also shown are the first and second derivatives of the fitting curve (second subplot), from which the slope information used in the first subplot is derived.

Figure 16. Logistic fit (S-shaped curve) of the regional network performance as a function of monitoring tower numbers.

The optimal foreground locations cover the main urban centers (i.e., Paris, Strasbourg, Nancy–Metz, Frankfurt, Bern–Zurich) as well as bordering regions of Germany and Switzerland, where large fossil fuel emissions originate (see Figs. 17a and 17c). Northern locations aim to capture large CO<sub>2</sub> plumes from the Benelux region, which strongly influence CO<sub>2</sub> spatial gradients over the domain, while the Ruhr Valley (north of Alsace) is covered by a specific measurement location. The domain center remains mostly devoid of ground-based measurement stations due to the absence of major plumes in the simulation. These areas have the lowest population densities and no significant industries or major highways.

For background concentrations, stations are located along the northern boundary and in the central part of the domain, with two additional sites identified in mountainous areas (i.e., the Black Forest and the Swiss Alps). Further experiments using simulated concentrations might enable a reduction in the number of background stations, though such decisions rely primarily on expert knowledge and spatial correlation, as discussed in Sect. 4.

Overall, 18 tower locations are required to constrain the major fossil fuel signals and their associated background concentrations over the simulation domain. The coordinates of the optimal ground-based measurement station locations (regional scale) and their performance values are provided in Table 2.

**Figure 17.** Resulting optimal centroids (black triangles) and their corresponding clusters for all images in the dataset for both the foreground and background networks (regional level) (a), location of the optimal centroids relative to the emission field (b), and the scoring matrix (c).

**Figure 18.** Binarized and inverted image of the score matrix used to mask the background tracer in the background analysis for the regional-scale application.

Table 2. Optimal results coordinates for regional-scale analysis.

|   | ]         | Foreground | I         | Background |          |           |
|---|-----------|------------|-----------|------------|----------|-----------|
|   | Longitude | Latitude   | Score (%) | Longitude  | Latitude | Score (%) |
| 1 | 2.2502    | 49.038     | 38.741    | 5.4000     | 48.682   | 49.213    |
| 2 | 3.8251    | 50.601     | 25.325    | 7.5282     | 48.353   | 46.954    |
| 3 | 6.2939    | 50.820     | 60.438    | 4.5487     | 50.025   | 52.224    |
| 4 | 6.6769    | 49.395     | 49.007    | 2.2076     | 49.998   | 51.608    |
| 5 | 7.6559    | 48.079     | 44.285    | 3.4846     | 48.737   | 48.323    |
| 6 | 8.1241    | 48.874     | 54.620    | 5.1446     | 47.284   | 50.103    |
| 7 | 8.3370    | 47.393     | 67.625    | 1.9948     | 47.722   | 43.669    |
| 8 | 8.5923    | 50.135     | 86.037    | 8.3795     | 47.201   | 45.174    |
| 9 | 8.9329    | 49.258     | 57.495    | 6.9323     | 50.190   | 49.829    |

#### 4 Discussions



As a result of the procedures carried out in this study, several aspects merit discussion: Among the preliminary considerations, the development of the optimal design scheme is based on two types of measurements (direct and indirect) and the measuring instruments themselves. The variety of instruments used in greenhouse gas monitoring requires that they be immersed in the gas flow to characterize their location relative to the measured concentrations. For this reason, CRO<sup>2</sup>A seeks to identify locations for ground-based measuring stations where greenhouse gas fluxes are expected to occur most frequently and for the longest observation periods. This approach is consistent with that of (Nalini et al., 2019), who prioritized the location of monitoring network stations over the magnitude of uncertainty reduction, as the former depends on prior and observational uncertainty values.

The primary purpose of the preprocessing stage is to reduce the volume of data to be processed and analyzed; this is one reason for implementing each of the applied transformations. Furthermore, this stage, along with these transformations, is a distinguishing feature of this study: in addition to automating data cleansing, the analysis does not require any statistical assumptions beyond those inherent to the model used.

No clustering algorithm can generally guarantee convergence to a global minimum. Achieving the optimal clustering often requires exploring a search space too large to exhaustively evaluate, unless certain problem-specific conditions are met. Since the global minimum lies within the set of solutions defined by the search space-whose size in this case is approximately proportional to k! (i.e., the factorial of the number of ground-based measurement stations in the monitoring network)-exploring a space of such magnitude entails high computational cost. The proposed algorithmic strategy in the processing stage addresses this by narrowing the search space (primarily during preprocessing), initializing starting points using information inherent in the data, and adopting a smart exploration method that diversifies candidate solutions to avoid local optima.




Like other algorithms, CRO<sup>2</sup>A explores only parts of the search space; its distinction lies in how these portions are selected and how the search is subsequently intensified in the post-processing stage.

Because different starting points produce different clusters whose general trends are then extracted, a selection strategy based on the marginal distributions of data point density was proposed.

The clustering procedure is performed using inverse weighting, taking the reciprocal of each point concentration as its weight. These weights identify key clustering points and allow systematic exclusion of noise from further analyses. An alternative is fuzzy clustering, implemented by setting specific constraints for each case. However, this method can be complex, potentially leading to a blurred boundary between objective and subjective outcomes. The optimal design scheme is structured around clustering algorithms such as k-medoids and k-means, which are fast, iterative, require relatively few iterations to converge, and have simple per-iteration calculations.

When the merging or splitting of two or more clusters is required, such actions should be approved by an expert, and the scheme rerun with the updated number of clusters. Future versions of CRO<sup>2</sup>A are expected to include an automatic strategy for this task.

As stated initially, this is a proposed solution within the framework of unsupervised machine learning. Accordingly, complementary validation of the results requires the judgment of a subject-matter expert, through which certain sensitive groups can be considered. The optimal design scheme is therefore intended as a decision-support tool for determining the locations of ground-based measurement stations.

Given the modular structure of CRO<sup>2</sup>A, additional variables of interest may be incorporated into monitoring network design; however, their inclusion must be assessed to ensure it does not bias the clustering results. For such evaluations, techniques such as principal component analysis and/or regularization are recommended to address data multidimensionality and to extract features from the most influential variables. This aspect was specifically considered during the development of the optimal design scheme, for which high-performance computing principles were applied to balance algorithmic complexity. This included managing dataset storage to avoid overloading system memory while reducing processing time, without compromising accuracy-which is more important than algorithm speed for an optimal design.

In all cases where CRO<sup>2</sup>A is employed, and depending on the user scientific objectives, networks of varying sizes are proposed to enable the separation of primary signals within the selected domain. Similarly, the specific features of this optimal design scheme are made available for the user discretion in applying them to a particular use case.

For brevity, a follow-up paper is proposed to compare the results of CRO<sup>2</sup>A with previously published findings, particularly those of (Ziehn et al., 2014a) and (Nickless et al., 2015). Furthermore, as part of the validation process, the designs generated by the proposed optimal design scheme will be compared with those derived from inverse modeling to demonstrate its competitiveness.

#### 5 Conclusions

This research developed an optimal design scheme, CRO<sup>2</sup>A, capable of supporting the deployment of greenhouse gas monitoring networks without the need for an inversion system, relying solely on direct atmospheric simulations.

The scheme employs an inversely weighted version of a modified clustering algorithm, combined with an optimization strategy that enables automated data analysis and processing to obtain essential information for designing atmospheric monitoring networks. The modifications applied to the classic *k*-medoid and *k*-means algorithms preserve their core characteristics while enhancing their ability to diversify and intensify the search within the solution space. CRO<sup>2</sup>A can generate optimal designs for both primary (foreground) and secondary (background) monitoring networks, accommodating the presence of pre-installed ground-based measurement stations.

This approach is computationally efficient and offers greater flexibility than existing network design tools, without employing a complex inversion system. It also avoids dependence on many of the typical inverse assumptions (e.g., *a priori* error statistics) inherent in inverse modeling studies. By clustering time series of atmospheric greenhouse gas concentration fields, the scheme ensures that ground-based measurement stations are sited where signals are most frequently present, according to the seasons or times of day defined by the user.

The CRO $^2$ A scheme converges rapidly, requiring relatively few iterations compared with the data size, owing to its initialization strategy. It retains the k-medoids and k-means property of low-complexity calculations per iteration, making it well suited for processing large datasets. An additional feature allows selection of the deployment area, masking potential locations outside predefined zones. Furthermore, CRO $^2$ A provides an effective tool for evaluating existing networks, identifying additional sites to expand them, or designing future observation networks without preexisting ground-based stations. The scheme can be used in conjunction with expert input to guide site selection and design iterations.

While the design of a monitoring network ultimately depends on specific objectives, CRO<sup>2</sup>A serves as a decision-support tool, providing key information to assist experts in deploying atmospheric monitoring networks across diverse landscapes and environments.

Code and data availability. Codes, data, and examples are publicly available at Matajira-Rueda et al. (2025) (https://doi.org/10.5281/zenodo. 17161303 and https://doi.org/10.5281/zenodo.17161462

## **Appendix A: Special features**

Several special features of the CRO<sup>2</sup>A optimal design scheme are briefly described below, which facilitate user analysis during processing for the design of atmospheric monitoring networks. As noted earlier, with CRO<sup>2</sup>A, monitoring network design and analysis can be performed separately in the foreground or background mode, or jointly in the so-called complete mode. Both analyses are executed sequentially and in an orderly manner, providing feedback on their respective results. In complete mode, the background mode is executed immediately after the foreground mode, propagating the results from one to the next.



Background measurement locations are estimated based on simulated large-scale background concentrations, while avoiding major sources and sinks of greenhouse gases. This process reduces the risk of false solutions in background mode for locations with considerably high greenhouse gas concentrations. This is achieved through a mask extracted from the foreground mode scoring matrix.

If the region under analysis contains pre-existing ground-based measurement stations, these may or may not be included in the monitoring network design (the latter being the default). When inclusion and batch processing are required (as illustrated in Fig. 11 to determine network performance as a function of the number of ground-based measurement stations), it is advisable to set the number of pre-existing stations as the initial value for the test interval.

For example, the results for the two applications presented in Sect. 3 are shown below, this time including the pre-existing stations in the ICOS network, listed in Table A1. For the urban-scale application, the test interval was set to [2,10], while for the regional-scale application, [12,15] was used, based on the information in Table A1.

# A1 Urban-scale application including ICOS network

Figure A1. Results for the urban-scale application considering the pre-existing ground-based measurement stations in the ICOS network, with a minimum number of stations k = 5 according to CRO<sup>2</sup>A.

#### A2 Regional-scale application including ICOS network

Comparing the main results of the two proposed applications with those presented in this section-which include ground monitoring stations already installed and part of the ICOS network-reveals how CRO<sup>2</sup>A attempts to assimilate such locations by incorporating them into the design. The most evident example is the regional-scale application, where, out of 12 possible pre-existing stations, 7 are effectively and completely assimilated (i.e., they coincide with the alternative network proposed by the optimal design scheme). In both case studies, when confronted with pre-existing stations exhibiting poor performance, CRO<sup>2</sup>A attempts to assimilate them.

If assimilation is not feasible, it incorporates these stations and generates complementary ones nearby to mitigate their impact on overall network performance. This approach is illustrated in the urban-scale application, where the scheme assimilates the

Figure A2. Results for the regional-scale application considering the pre-existing ground-based measurement stations in the ICOS network, with a minimum number of stations k = 13 according to CRO<sup>2</sup>A.

**Table A1.** Pre-existing ground-based measurement stations belonging to the ICOS network, considered in the analysis at urban or regional scale.

| Scale          | Longitude | Latitude | Reference |
|----------------|-----------|----------|-----------|
| Regional       | 8.1755    | 47.189   | BRM       |
| Regional       | 8.6750    | 49.417   | HEI       |
| Regional       | 8.4249    | 49.091   | KIT       |
| Regional       | 8.3973    | 47.482   | LHW       |
| Urban/Regional | 5.5036    | 48.562   | OPE       |
| Regional       | 2.1420    | 48.723   | SAC       |
| Regional       | 7.9166    | 47.917   | SSL       |
| Regional       | 2.1125    | 47.965   | TRN       |
| Urban/Regional | 4.0611    | 49.243   | MDH       |
| Regional       | 2.4205    | 49.005   | GNS       |
| Regional       | 2.3018    | 49.012   | AND       |
| Regional       | 3.9747    | 49.236   | ORM       |

two existing towers, aiming to preserve the original design of Fig. 14d with three stations, while significantly modifying the location of one (the southeasternmost station in the city of Fig. A1a). It should be noted that in both the urban-scale and regional-scale applications, CRO<sup>2</sup>A strives to include existing stations wherever possible. However, when such stations exhibit low performance, the overall quality of the network is inevitably compromised (e.g., for Reims, performance decreases from 43.41% to 32.40%, and for Grand Est, from 53.73% to 47.69%).

Similarly, the design can be performed by preloading potential locations for ground-based measurement stations, as in Ziehn et al. (2014b), yielding an optimal network under these conditions. Continuing with the analysis of pre-existing stations in the study region, the current monitoring network can also be evaluated, obtaining scores for each station and an approximate





overall network score. This evaluation is performed using the metrics proposed in this study, which form the basis of the CRO<sup>2</sup>A optimality criterion.

Regarding the masking process, CRO<sup>2</sup>A can automatically generate a mask (with some resolution limitations), as illustrated in Fig. 5b; alternatively, the user can load a custom mask according to the objectives of the analysis. In the same vein, CRO<sup>2</sup>A includes an option called "Special Mask", which enables masking of a main field using another field (e.g., it can be used to mask an anthropogenic field using a biogenic field). Unlike other masking processes, where a single mask is applied to all images of the analyzed field in foreground mode, this option performs masking on an image-by-image basis between the two selected fields.

CRO<sup>2</sup>A also allows the selection of a rectangular subregion within the area defined by the input data, graphically or textually, by specifying the minimum and maximum latitude and longitude values. Although this approach avoids recompiling the input data, it remains subject to limitations in relation to the minimum resolution required for adequate cluster analysis.

# Appendix B: Technical notes and definitions

The core of the proposed design scheme is cluster analysis, which aims to identify characteristic groups formed through the recognition of specific patterns, thereby enabling the extraction of useful information on similarity or dissimilarity from the analyzed data (Doan et al., 2022).

Among the various clustering algorithms within pattern recognition, two general cases exist: supervised and unsupervised. Their use depends on whether the training class label is available (de Burgh-Day and Leeuwenburg, 2023). Given the characteristics of the application described in this document, special attention is given to the unsupervised case due to the uncertainty of the output for a given input, i.e., the absence of a reference or expected ideal result (ground truth). The proposed design scheme incorporates elements of clustering algorithms such as *k*-means, *k*-medoids, and the hierarchical variant, which help constrain the set of solutions. These are combined using an inversely weighted clustering strategy to improve overall performance without significantly increasing algorithmic complexity.

It should be noted that in the proposed design scheme, the number of *k* clusters characterizing the data varies, and the process involves adjusting the centroid positions of these *k*-clusters according to the mean related to the density and inverse weighting of the same data.

In addition to preparing the data for further processing, the clustering analysis in the proposed design scheme aims to extract the maximum possible information on the behavioral trends of simulated GHG concentrations over a region by employing techniques from various scientific fields to explore and exploit similarities and dissimilarities in the data.

The proposed design scheme used for this analysis is named Designer of Optimal Atmospheric Observation Networks (CRO<sup>2</sup>A; French acronym for *Concepteur de Réseaux Optimal d'Observation Atmosphérique*).

The development of CRO<sup>2</sup>A followed the basic steps of a clustering task, supplemented by various methods to enhance data analysis:

### Figure B1. CRO2A





- **Selection of features:** Based on their relative significance, the variables selected as features are the latitude, longitude, and GHG concentration at each point in the analysis region. During clustering, variables are indirectly processed through their projections onto the *x-y* plane, while retaining the nominal and ordinal properties of interval scale features, i.e., features for which the ratio between two values is meaningless but the difference between them is meaningful. This selection encodes the information while minimizing redundancy, depending on preprocessing, which is an essential stage involving data cleaning, filtering, and masking.
- Selection of the membership measure: This measure employs a function to model the dissimilarities (or similarities) between data points. Its variables are directly related to the selected features. In this study, a widely accepted proximity measure is used, as shown in (B1).

$$f_{sd}(\mathbf{p}, \mathbf{c}) = \left(\sum_{i=1}^{m} \sum_{j=1}^{n} \mathbf{w}_{j} \|\mathbf{p}_{j} - \mathbf{c}_{i}\|^{2}\right)^{1/2},$$
 (B1)

where  $f_{sd}$  is the membership function (for dissimilarity modeling) for r=2 from the  $L^r$ -norm (Euclidean distance measure). Here, n is the number of data points, m is the number of clusters,  $p_j$  is the j-th data point, and  $c_i$  is the centroid of the i-th cluster. Both  $p_j$  and  $c_i$  correspond to an (x,y) coordinate pair derived from the width and height values of the j-th image under testing in the x-y plane. Lastly,  $w_j \geq 0$  is the weight of  $p_j$  according to a scoring matrix. The membership function is used to model data patterns, minimizing the dissimilarity threshold (or maximizing the similarity threshold, depending on the approach), thereby optimizing both the quantity and quality of the resulting clusters.

- Selection of clustering criterion: This refers to the formation of clusters, which is not limited to compact groups, as the data may exhibit both low and high dispersion. The clustering criterion is therefore linked to network quality based on available data. Network quality is defined using a scoring matrix (explained in Sect. 2.2.1) that rates each point in the analyzed region according to the frequency of relatively high concentrations. A trend analysis is implemented via the basic sequential algorithmic scheme (BSAS) to address the recurring question of the optimal number of clusters (i.e., number of towers and their locations in the region under analysis). Accordingly, the clustering criterion is expressed in terms of the trend in the ratio between the total number of ground-based measurement stations and their quality.






- Selection of clustering algorithms: The proposed design scheme combines k-means, k-medoids, and hierarchical clustering strategies to uncover implicit groups in the dataset. Depending on user needs or problem characteristics, the algorithm may run in either k-medoids mode (default) or k-means mode, both always supported by hierarchical clustering.
- Validation of results: The clustering results-specifically, the number and locations of ground-based measurement stationsare evaluated using parametric and non-parametric tests based on the previously defined clustering criterion, with reference to the region scoring matrix.
- **Interpretation of results:** Since no ground truth is available, validation and evaluation must be performed by field experts through additional experimental tests, ensuring that results have practical relevance. The design scheme also offers alternatives to the primary design (of equal or slightly lower quality) if physical implementation of the main design is not feasible.
- Clustering tendency: This refers to tests designed to determine whether an inherent clustering structure exists in the data. The hierarchical clustering algorithm is the primary clustering tendency test in this study, applied to assess trends in the quality and number of ground-based measurement stations in the designed network.

## **Appendix C: Clustering algorithms**

To formally define the clustering process, let  $\mathfrak{C} = \{\mathfrak{c}_1, \mathfrak{c}_2, \mathfrak{c}_3, \dots, \mathfrak{c}_{\mathfrak{m}}\}$  be the m-clustering of  $\mathfrak{D} = \{d_1, d_2, d_3, \dots, d_n\}$ , representing the dataset of n data points partitioned into m subsets, where  $d_n = [d_{n,1}, d_{n,2}, d_{n,3}, \dots, d_{n,f}]^{\mathsf{T}}$  is the n-th data point in the f-dimensional space. Through a membership function, it is possible to quantify similarity or dissimilarity using the numerical values corresponding to each of the f features stored in the data points. The correspondence between similarity and dissimilarity measures allows the analysis to be performed interchangeably from either perspective. It is common to refer to the degree of similarity (or dissimilarity) as a distance measure. A widely used membership function models the proximity (distance) between two points, providing a mathematical basis for this correspondence:

$$f_{L^r}(\boldsymbol{x}, \boldsymbol{y}) = \left(\sum_{i=1}^n \|\boldsymbol{x}_i - \boldsymbol{y}_i\|^r\right)^{1/r},\tag{C1}$$

where  $f_{L^r}$  is the similarity-dissimilarity modeling function known as the  $L^r$ -norm, with i = 1, 2, 3, ..., n, where n is the number of data points. Each i-th data point is defined by the values  $x_i$  and  $y_i$  of x and y, respectively. In  $f_{L^r}$ , r > 0; in particular, for  $r = 1, 2, \infty$ , the Manhattan  $(L^1)$ , Euclidean  $(L^2)$ , and maximum  $(L^\infty)$  norms are obtained, respectively.

Before describing in the following section the characteristics of the optimization algorithm, it is important to note that an appropriate m-clustering  $\mathfrak C$  consists of m subsets containing elements of  $\mathfrak D$  such that elements with similar features belong to the same cluster, whereas elements with dissimilar features belong to different clusters. In addition, the m-clustering  $\mathfrak C$  must satisfy the following conditions:

Condition 1.  $\mathfrak{c}_k \neq \emptyset, k = 1, 2, 3, \dots, m$ .

Condition 2.  $\cup_{k=1}^m \mathfrak{c}_k = \mathfrak{C}$ .

**Condition 3.**  $c_{k_1} \cap c_{k_2} = \emptyset$ ,  $k_1 \neq k_2$ , and  $k_{1,2} = 1, 2, 3, ..., m$ .

# 610 Appendix D: Additional tests

Presented below are additional results complementing those in Sect. 3, specifically for the urban-scale application described in Sect. 3.1.

Figure D1. Score percentages per tower and overall network performance value for the urban-scale application with k = 4: (a) performance scores and (b) corresponding tower locations.

Author contributions. DMR developed the optimal design scheme and performed various tests using datasets provided by CA. CA performed model simulations, validated the projected results, and contributed to the analysis of the optimal design scheme outcomes. TL provided the original project concept and performed analysis and validation of the results. DMR, CA, and TL contributed extensively to writing this document.

Competing interests. The authors declare that they have no competing interests.

Acknowledgements. This study was supported by the Postdoctoral Research Program of the Université de Reims Champagne-Ardenne (URCA) and the French Ministry of Research and Education (MESRI) through the Chaire de Professeur Junior (CASAL project). Part of this study was funded by the National Center for Scientific Research (CNRS), the French National Space Agency (CNES), and the European Space Agency (ESA) as part of the MAGIC aircraft program. We thank the ROMEO HPC computing facility of URCA for enabling the algorithm testing.

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
