# Peer review of "A novel cluster-based learning scheme to design optimal networks for atmospheric greenhouse gas monitoring (CRO2A version 1.0)"

_EGUsphere, 2025_

## Author Comment (AC1)

**Reviewer:** *Matajira-Rueda et al. present a novel approach to the optimization of new ground-based stations in a greenhouse gas observation network. Many previous approaches have relied on the inverse modelling methodology which is traditionally used to optimize for the flux estimates using concentration measurements from these ground-based network stations and prior information. This has computational challenges, as the optimization requires running components of the inversion which require extremely large datasets, and having to repeat this a large number of times in order to determine which set of stations achieves the best result with respect to some objective function, usually related to uncertainty reduction. The approach presented by the authors in this paper propose a machine-learning approach which is based on identification of clusters in the region, and then optimizing the location of sites which observe these clusters. Approaches are implemented to reduce the dimensionality of the data to improve on the computational time for running the algorithm, which therefore allows for more repeats of the process to be undertaken with different starting values to ensure that the optimal solution is achieved, rather than a local optimum.*

*The authors present the approach is a logical and clear manner, and clearly describe each step. The manuscript is easy to follow, even if no prior knowledge of inversions or machine learning. The figures and tables complement the explanation of the method and discussion of results.*

*I think that the manuscript is sufficient in it's current form to present the proposed method and application.*

*I think it may be worth emphasizing that regardless of which method is used for optimizing the location of measurement stations, there is still a requirement for a thorough understanding of the transport model/models that will be used to generate the simulated concentrations, as locations where these models are known to perform poorly should be excluded from the search space. While the inverse modelling approach may not be used for determining the optimal network, the resulting network still needs to be compatible with the approach and take into account the challenges that need to be dealt with during the inversion procedure in order to achieve estimates of the posterior fluxes. For example, there needs to be an appreciation for the prior information that will be provided for the inversion, as the ultimate aim will be to ingest the concentration data from the observation network, together with the prior information, to provide estimates of fluxes. Locations that are heavily influenced by regions where the prior information is poor or highly uncertain can be problematic, as even if a new measurement station in that region contributes towards uncertainty reduction, the resulting posterior uncertainty is still very high, particularly if this is combined with error in the atmospheric transport model for that region. Approaches that use uncertainty reduction as the basis for objective function of the network design can penalize regions such as these by manipulating the uncertainty in*

*these regions so that the optimization solutions with stations which see these locations do not overly dominate at the cost of seeing other regions which new stations can better contribute towards characterizing. Regions with high uncertainty are also those regions with high concentrations normally, so I think that both approaches would try to find solutions that view the same regions. The exception is CO2, as during periods when photosynthesis dominates, the concentrations in the surrounding regions influenced by air masses passing over these regions may have concentrations that are pulled lower, but actually the uncertainty in the models that describe photosynthesis can be very high, so if the objective was to improve on the prior fluxes for these regions, it would still be desirable to have stations that viewed these regions in the network. Therefore, there may need to be some adaptations to the method to account for large negative fluxes, or when regions have both large negative fluxes and anthropogenic fossil fuel contributions.*

*I'd certainly be interested to see how this method compares to the previous inverse modelling based approaches if both are provided with the same inputs.*

*Specific comments:*
*I think some clarifications in the caption would assist to allow the figures and tables be more stand-alone.*
*Figure 11, 13, 16: It's not clear what is the y-axis of the lower figure. Line 324-333*
*Figure 14, 17: the caption does not explain what's in (d).*
*Table 2. It's not clear from the title or row labels why there are 9 rows, or what the order signifies, if anything.*

***Citation***: *https://doi.org/10.5194/egusphere-2025-4112-RC1*
* * *
The authors appreciate your time and willingness to review the proposed article. Following your comments and instructions, we have made the indicated and highlighted modifications in a new version of the document.

We also appreciate your comments and suggestions, which we value as they motivate us to continue exploring diverse application scenarios for CRO$^2$A, as well as its possible future updates.

As you mention, most approaches focus on inverse modeling, and as is well known, its computational cost can be very high. Furthermore, it relies on information processed through Bayesian probabilistic assumptions. What we propose with CRO$^2$A is the exploration of a different perspective from this traditional one, using a metric other than uncertainty reduction. The exploration is done automatically based on characteristics inherent to pattern recognition. The formulation of a different objective function allows

both the exploration and exploitation of the solution space, in turn increasing convergence toward a global optimum.

We fully agree with you on the importance of the atmospheric transport model used and its appropriate parameter settings. The quality of $CRO^2A$'s results depends directly on the quality of the simulated data in the corresponding atmospheric transport model. Therefore, based on $CRO^2A$'s performance using data from other transport models (different from WRF and CAMS), we hope to enable their use in future versions to leverage the advantages of each.

Although we indirectly influence uncertainty, $CRO^2A$ focuses more on the trend of concentration behavior over time and space, taking advantage of automated analysis that, in turn, uses descriptive statistics to characterize them. The constraints applied to the data during processing are systematic, allowing the algorithm to learn from them without relying on assumptions that could bias the results.
We would like to highlight two of your comments, which precisely reveal some weaknesses in the inverse modeling design and definitely create an opportunity to test alternative solution strategies with $CRO^2A$:

Firstly, regarding those locations heavily influenced by regions where prior information is deficient or highly uncertain, we believe that through a set of simulations using different flux fields as input, could be tested in a later version of $CRO^2A$. Except that the transport model realizations could have a significant computational cost.

Secondly, regarding the penalty imposed on regions by approaches based on uncertainty reduction, we suggest a way to avoid this penalty by filling the "gaps" with the information provided by $CRO^2A$, after using a set of realizations of the transport model (same flow field), so that transport errors are taken into account. The main limitation is the cost of running ensemble simulations (for both transport and emissions errors).

We also find your perspective on the inclusion of biogenic fields very interesting. We are well aware of this aspect and consider it undeniably necessary. Therefore, these fields have been under observation since the beginning of development, but the results are still being evaluated. The main difficulty lies in the "smoothness" or "flatness" of the biogenic fields. Our key to continuing research in this direction is the use of a complementary technique, which, at first glance, may be linked to a segmentation process based on the sets of species to be analyzed.

Finally, we would like to inform you that we already have comparative results with the Australian network presented in:

*ZIEHN, T., et al. Greenhouse gas network design using backward Lagrangian particle dispersion modelling– Part 1: Methodology and Australian test case. Atmospheric Chemistry and Physics, 2014, vol. 14, no. 17, pp. 9363-9378.*

These results have already been appended to this article (Lines 636-691). We are also conducting tests on the African network presented in:

NICKLESS, Alecia, et al. Greenhouse gas network design using backward Lagrangian particle dispersion modelling–Part 2: Sensitivity analyses and South African test case. Atmospheric Chemistry and Physics, 2015, vol. 15, no. 4, pp. 2051-2069.

However, these latter results will be published later.

About "specific comments":
**Figure 11, 13, 16: It's not clear what is the y-axis of the lower figure. Line 324-333**
To clarify, we have added the following paragraph on the indicated lines:

Line 333-338
The logistic function representing the fitted model and its first two derivatives are used to calculate the optimal threshold (see the first and second sub-Fig. 11, respectively) according to the procedure proposed by McDowall and Dampney (2006), which makes use of intersections between certain straight lines (including the slope line at the midpoint of the logistic curve, obtained by means of the derivative in the second sub-Fig. 11) to calculate the baroreflex threshold and saturation points.
It should be noted that the performance represented is normalized; therefore, the vertical axes of this figure and its first two derivatives, shown in Fig. 11, are dimensionless.

**Figure 14, 17: the caption does not explain what's in (d).**
The figure captions have been corrected as shown below:

Figure 14. Resulting optimal centroids (black triangles) and their corresponding clusters for all images in the dataset for both the foreground (a) and background (b) networks (urban level), location of the optimal centroids relative to the emission field (c), and according to the scoring matrix (d).

Figure 17. Resulting optimal centroids (black triangles) and their corresponding clusters for all images in the dataset for both the foreground (a) and background (b) networks (regional level), location of the optimal centroids relative to the emission field (c), and according to the scoring matrix (d).

***Table 2. It's not clear from the title or row labels why there are 9 rows, or what the order signifies, if anything.***

The numbering in Table 1. and Table 2. has been removed as it had no relevant meaning other than to list the monitoring stations obtained.

Furthermore, the caption of both tables has been modified as follows:

"Table 1. Optimal results coordinates for urban-scale analysis according to Fig. 14. ==Two columns are presented: the first (Foreground) for the main monitoring network and the second for the background network. Both networks contain three ground monitoring stations, since they are designed as one-to-one networks and because 3 is the minimum value (threshold in Fig. 13) obtained from the analysis.==",

"Table 2. Optimal results coordinates for regional-scale analysis according to Fig. 17. ==Two columns are presented: the first (Foreground) for the main monitoring network and the second for the background network. Both networks contain nine ground monitoring stations, since they are designed as one-to-one networks and because 9 is the minimum value (threshold in Fig. 16) obtained from the analysis.==",

respectively.

Thank you in advance for your attention and collaboration.

Sincerely,

**David Matajira-Rueda**
**Charbel Abdallah**
**Thomas Lauvaux**

Additional :
Following your observations, the authors have included some words or comments to improve the understanding of the document; therefore, below we list the lines in which such inclusions are found:
Line 16
Lines 113-114
Line 184
Line 189
Line 326
Lines 344-347
Lines 435-436